# Comparative diagnostic accuracy between simplified and original flow cytometric gating strategies for peripheral blood neutrophil myeloperoxidase expression in ruling out myelodysplastic syndromes

Tatiana Raskovalova[1,2]*, Laura Scheffen[1], Marie-Christine Jacob[1,2], Claire Vettier[3], Bénédicte Bulabois[3], Gautier Szymanski[3], Simon Chevalier[3], Nicolas Gonnet[4], Sophie Park[1,5], José Labarère[6,7]

1 Institute for Advanced Biosciences, INSERM U1209, CNRS UMR 5309, Université Grenoble Alpes, Grenoble, France, 2 Laboratoire d'Immunologie, Grenoble University Hospital, Grenoble, France, 3 Laboratoire d'Hématologie Biologique, Grenoble University Hospital, Grenoble, France, 4 CIC 1406, INSERM, Université Grenoble Alpes, Grenoble University Hospital, Grenoble, France, 5 Clinique Universitaire d'Hématologie, Grenoble University Hospital, Grenoble, France, 6 Clinical Epidemiology Unit, Grenoble University Hospital, Grenoble, France, 7 TIMC-IMAG, UMR 5525, CNRS, Université Grenoble Alpes, Grenoble, France

* TRaskovalova@chu-grenoble.fr

## Abstract

### Background

Flow cytometric analysis of peripheral blood neutrophil myeloperoxidase expression is accurate in ruling out myelodyplastic syndromes (MDS) but might not be suitable for implementation in busy clinical laboratories. We aimed to simplify the original gating strategy and examine its accuracy.

### Methods

Using the individual data from 62 consecutive participants enrolled in a prospective validation study, we assessed the agreement in intra-individual robust coefficient of variation (RCV) of peripheral blood neutrophil myeloperoxidase expression and compared diagnostic accuracy between the simplified and original gating strategies.

### Results

Cytomorphological evaluation of bone marrow aspirate confirmed MDS in 23 patients (prevalence, 37%), unconfirmed MDS in 32 patients (52%), and was uninterpretable in 7 patients (11%). Median intra-individual RCV for simplified and original gating strategies were 30.7% (range, 24.7–54.4) and 30.6% (range, 24.7–54.1), with intra-class correlation coefficient quantifying absolute agreement equal to 1.00 (95% confidence interval [CI], 0.99 to 1.00). The areas under the receiver operating characteristic (ROC) curves were 0.93 (95% CI, 0.82–0.98) and 0.92 (95% CI, 0.82–0.98), respectively ($P$ = .32). Using simplified or original

**Data Availability Statement:** All relevant data are within the paper and its Supporting Information files.

**Funding:** Becton Dickinson Bioscience provided antibodies free of charge (Dr Raskovalova). Statistical analysis was performed within the Grenoble Alpes Data Institute (ANR-15-IDEX-02). (José Labarère). This research received no other specific grant from any funding agency in the public, commercial, or not-for-profit sectors. The funders had no role in study design, data collection and analysis, decision to publish, or preparation of the manuscript.

**Competing interests:** The authors have declared that no competing interests exist.

gating strategy, intra-individual RCV values lower than a pre-specified threshold of 30.0% ruled out MDS for 35% (19 of 55) patients, with both sensitivity and negative predictive value estimates of 100%.

## Conclusions

The simplified gating strategy performs as well as the original one for ruling out MDS and has the potential to save time and reduce resource utilization. Yet, prospective validation of the simplified gating strategy is warranted before its adoption in routine.

## Trial registration

ClinicalTrials.gov Identifier: NCT03363399 (First posted on December 6, 2017).

## Introduction

Myelodysplastic syndromes (MDS) are clonal bone marrow neoplasms predominating in the elderly and characterized by dysplasia and ineffective hematopoiesis leading to peripheral blood cytopenias [1]. The diagnostic work-up of MDS relies on cytomorphological evaluation of bone marrow, which may be complemented by conventional cytogenetic, flow cytometry, and molecular analysis by next generation sequencing techniques [2].

Suspicion of MDS is one of the most common reasons for bone marrow aspiration in older patients with persistent peripheral blood cytopenias of unclear etiology. Because of the relatively low prevalence of disease among subjects referred for suspected MDS [3], many patients are exposed to unnecessary bone marrow aspiration-related discomfort and harms. Bone marrow aspiration and biopsy are invasive procedures, with 20% of patients reporting a moderate level of pain for 7 days or more [4–6]. Although infrequent, procedure-related complications (hemorrhage and infection) may be associated with significant morbidity or even be life-threatening [7]. In this context, a valid and reliable assay based on peripheral blood sample that accurately discriminates MDS from other cytopenia etiologies might exclude MDS without requiring invasive bone marrow aspiration [8, 9].

Myeloperoxidase (MPO) is an enzyme synthetized during myeloid differentiation and constitutes the major component of neutrophil azurophilic granules [10]. MPO cytoplasmic expression is associated with degranulation of mature granulocytes [11], a classical dysplastic feature of MDS [12]. Cytoplasmic expression of MPO can be analyzed using various approaches, including immunocytochemical staining and flow cytometry.

The accuracy of peripheral blood neutrophil MPO expression quantified by flow cytometric analysis for ruling out MDS is supported by three primary studies totaling 211 individuals [13, 14]. Although promising, the gating strategy had practical limitations for routine application in busy clinical flow cytometry laboratories because of its complexity. The Boolean combination of gates was time consuming and relied on the expertise of the operators for individualizing cell subpopulations (neutrophils, eosinophils, lymphocytes, and monocytes), before measuring neutrophil MPO expression [13]. Simplification of the original gating strategy would have the potential to save time and resource, without compromising diagnostic accuracy.

The purpose of the present study was to refine the original gating strategy for quantifying peripheral blood neutrophil MPO expression and to examine the accuracy of the resulting

simplified gating strategy in ruling out myelodysplastic syndromes. More specifically, we aimed to quantify the agreement and comparative accuracy between the simplified and original flow cytometric gating strategies in estimating intra-individual robust coefficient of variation (RCV) of peripheral blood neutrophil MPO expression among consecutive unselected patients referred for suspected MDS.

## Methods

### Study design

We used the individual patient data from a prospective validation study of the original gating strategy by comparison with a reference standard in consecutive unselected patients. Descriptions of the design and primary outcomes of this study have been reported elsewhere [14].

### Participants

Sixty-two consecutive eligible adults were recruited at a single university-affiliated hospital in France, between February and September 2018 [14]. Patients aged 50 years and older were eligible if they were referred for suspected MDS, based on medical history and peripheral blood cytopenia. Given the very low prevalence of myelodysplastic syndromes below the age of 50 years [15], enrolling younger participants would have produced imprecise estimates for this subgroup of patients. Peripheral blood cytopenia was defined by hemoglobin concentration $<10$ g/dL, platelet count $<100 \times 10^9$/L, and/or absolute neutrophil count $<1.8 \times 10^9$/L [16]. We used individual data obtained from patients with milder level of peripheral blood cytopenia, in order to examine the robustness of our findings as part of a validation study.

### Flow cytometric analysis

**Blood collection.** Peripheral blood samples were collected in BD Vacutainer® 5 ml K2E (EDTA) anticoagulant plastic tubes (Ref 368861, BD Diagnostics, Le Pont de Claix Cedex, France). They were processed on the same day of collection or stored at 4˚C overnight and processed within 24 h of collection.

**Immunocytochemical staining.** Aliquots of whole blood sample (50 μL) were stained for 15 min at room temperature with cell surface markers (-PerCP-Cy5.5 [clone HI98], CD11b-APC [clone D12], CD16-APC-H7 [clone 3G8], CD14-V450 [clone MΦP9], and CD45-V500 [clone HI30]) [13].

The fixation and permeabilization phases were performed using the BD IntraSure™ kit (BD Biosciences, San Jose, CA, USA) in three steps, with incubation at room temperature in the dark, according to manufacturer recommendations as follows. The first step was the fixation of cell surface markers (adding 100 μL of Reagent A; vortex mixing; and incubating for 5 min). The second step was the lysis of red blood cells and decantation (adding 2 mL of 1X BD FACS Lysing Solution, vortex mixing; incubating for 10 min; centrifugating at 800 g for 5 minutes; and decanting the supernatant). The third step was permeabilization and staining for intracellular marker (adding 50 μL of Reagent B; vortex mixing; adding 10 μL MPO-PE (clone 5B8); vortex mixing; incubating for 10 min; centrifugating at 800 g for 5 minutes; decanting the supernatant; resuspending the cell pellet in 2 mL of PBS; centrifugating at 800 g for 5 minutes; decanting the supernatant; resuspending the cell pellet in 0.5 mL of PBS).

Technical information for reagents used for immunocytochemical staining is presented in S1 Table. All antibodies, BD FACS™ Lysing Solution (BD Biosciences, San Jose, CA, USA) and BD IntraSure™ kit were obtained from BD Biosciences (San Jose, CA, USA).

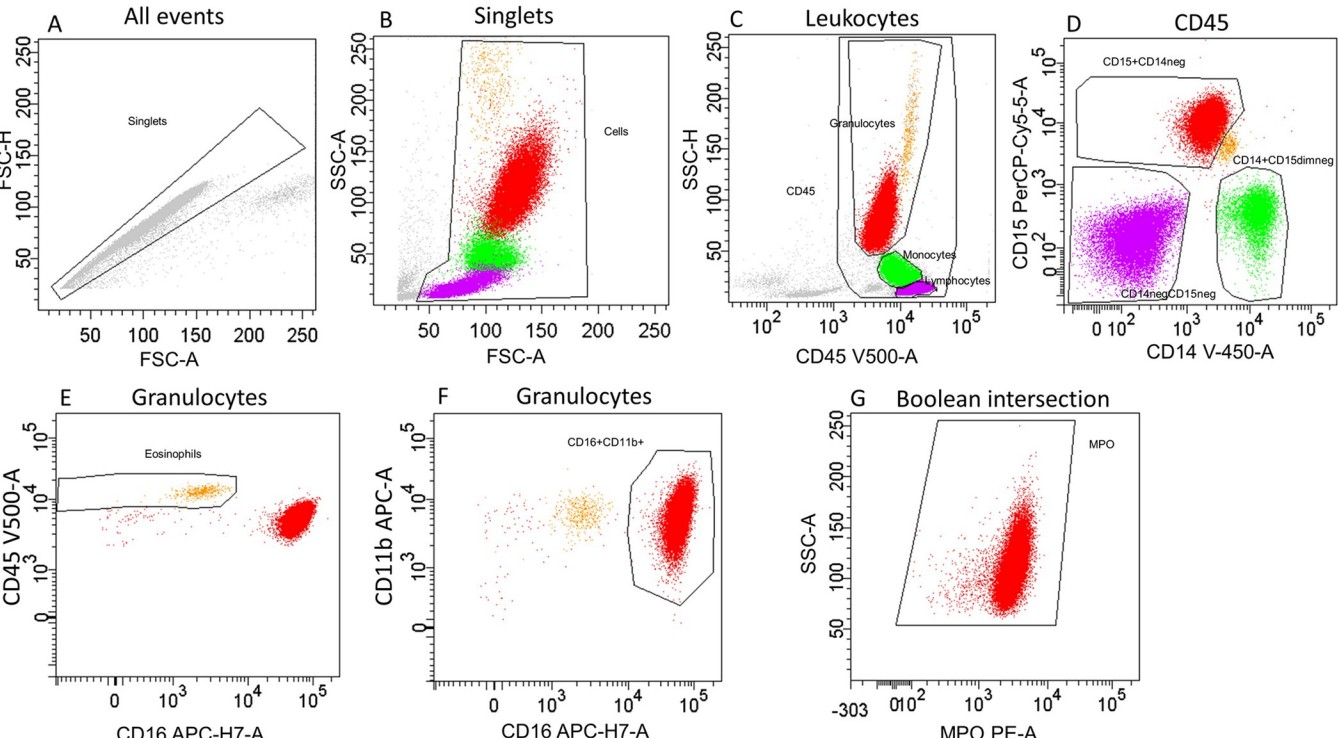

**Fig 1. Original flow cytometric gating strategy for quantifying peripheral blood neutrophil myeloperoxidase expression.** CD45+ cells were first individualized by crossing the singlet gate (A), FSC-SSC leukocytes (B), and CD45-positive gate (C). Three populations including granulocytes (CD15+ CD14-), monocytes (CD14+ CD15dim/-), and lymphocytes (CD15- CD14-) were identified (D). Eosinophils were individualized by CD45high CD16 low (E). Mature neutrophils were visualized by [CD15+ CD14-] [CD45low CD16 high] [CD16+ CD11b+] and selected by Boolean intersection: [CD15+ CD14-] [CD16 + CD11b+] with exclusion of [CD45high CD16 low] [CD14+ CD15dim/-] [CD15- CD14-] (F). RCV for MPO was estimated on the MPO gate conditioned on all visualized granulocytes (red dots) without threshold (G). The populations identified were lymphocytes (purple), monocytes (green), eosinophils (orange), MPO mature neutrophils (red). Abbreviations: CD = cluster of differentiation; FSC-A = forward scatter area; FSC-H = forward scatter height; MPO = myeloperoxidase; RCV = robust coefficient of variation; SSC-A = side scatter area; SSC-H = side scatter height.

**Cell acquisition and analysis.** At least 10,000 neutrophils were acquired on a three-laser, eight-color BD FACSCanto-II[TM] flow cytometer (BD Biosciences, San José, CA, USA) and analyzed using BD FACSDiva Software. Overview of lasers for FACSCanto-II TM is presented in S2 Table.

**Original gating strategy.** The original gating strategy was based on Boolean combination of gates, using "AND", "OR", and "NOT" logic, in order to individualize cell subpopulations (negative selection of granulocytes) (Fig 1). Briefly, CD45+ cells were first individualized by crossing the singlet gate (A) according to [17], FSC-SSC leukocytes (B), and CD45-positive gate (C). Three cell populations, including granulocytes (CD15+ CD14-), monocytes (CD14 + CD15dim/-), and lymphocytes (CD15- CD14-), were identified (D). Eosinophils were individualized by CD45high CD16 low (E). Mature neutrophils were visualized by [CD15+ CD14-] [CD45low CD16 high] [CD16+ CD11b+] and selected by the following Boolean intersection: [CD15+ CD14-] [CD16+ CD11b+] after excluding [CD45high CD16 low] [CD14 + CD15dim/-] [CD15- CD14-] (F). CD15 is considered a pan-granulocytic phenotypic marker [10, 18, 19] while CD14 was used as a monocyte phenotypic marker [10, 19, 20]. CD16, a marker expressed by neutrophils, was used to exclude eosinophils (CD16 low). CD11b is a classical human neutrophil phenotypic marker [10, 18, 19, 21]. RCV for MPO was estimated on the MPO gate conditioned on all visualised granulocytes (red dots) without threshold (G).

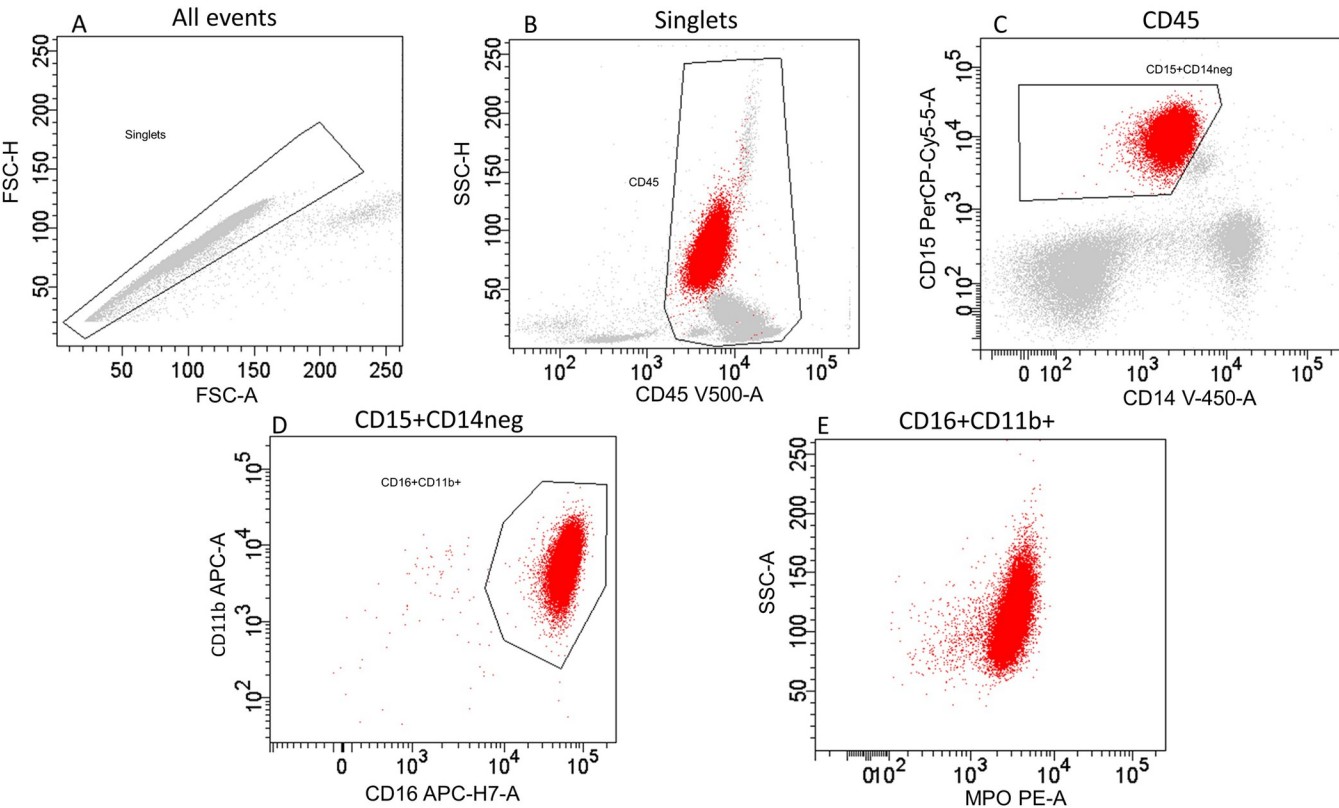

**Fig 2. Simplified flow cytometric gating strategy for quantifying peripheral blood neutrophil myeloperoxidase expression.** CD45+ cells were first individualized by crossing the singlet gate (A) and CD45 positive gate (B). Population of granulocytes (CD15+ CD14-) was identified (C). Gated cells were selected based on CD16/CD11b double positivity (D) The MPO SSC dot plot (E) was used only to visualize MPO expression of mature neutrophils. The cell population identified was MPO mature neutrophils (red dots). Abbreviations: CD = cluster of differentiation; FSC-A = forward scatter area; FSC-H = forward scatter height; MPO = myeloperoxidase; RCV = robust coefficient of variation; SSC-A = side scatter area; SSC-H = side scatter height.

Although rectangular gating is more straightforward to implement and easier to learn for operators [22], we used manually placed polygonal gates as described in a previous study [13].

**Simplified gating strategy.** The simplified gating strategy was hierarchical (positive selection of granulocytes), obviating Boolean operators and reducing the number of gates from 12 to 4 and the number of dot plots from 7 to 5 (Fig 2). Briefly, CD45+ cells were first individualized by crossing the singlet gate (A) and CD45 positive gate (B). Population of granulocytes (CD15+ CD14-) was identified (C). Mature neutrophils were individualized (CD16+ CD11b+) and gated cells were selected based on CD16/CD11b double positivity (D). The MPO SSC dot plot (E) was used only to visualize MPO expression of mature neutrophils. FSC-SSC leukocytes gate, populations of monocytes, lymphocytes and eosinophils were not included as part of this analysis. The original and simplified gating strategies were both performed by operators who were blinded to the reference diagnosis.

## Myeloperoxidase expression

MPO expression in neutrophil population for peripheral blood sample obtained from a given individual was expressed as intra-individual RCV [13]. The intra-individual RCV was calculated as the robust standard deviation divided by the median fluorescence intensity. The robust standard deviation is a function of the deviation of data points to the median of the study population [13]. RCV is robust to outliers and constitutes an alternative to the coefficient of

variation for skewed distributions. Intra-individual RCV was expressed as percentage and reflected the variability in MPO expression in the peripheral blood neutrophil population for an individual.

## Standardized instrument settings

We complied with the FranceFlow group standard operating procedure in order to standardize instrument settings [23]. The voltage for each photomultiplier tube was set to reach the target MFI of the FranceFlow-validated lot of Rainbow beads (target MFI ± 2%). Fluorescence compensation was calculated using CompBeads (BD Biosciences, San Jose, CA, USA) with BD FACSDiva Software (BD Biosciences, San Jose, CA, USA). For instrument performance monitoring, CS&T IVD Beads (BD Biosciences, San Jose, CA, USA) were run daily. Rainbow calibration particles (BD Sphero[TM], BD Biosciences, San Jose, CA, USA) were analyzed daily and photomultiplier tubes were adjusted if needed (target MFI ± 15%) [24].

## Reference standard

The reference diagnosis of MDS was established according to current guidelines [16]. Cellular morphology and percentage of excess blasts in bone marrow were evaluated by experienced hematopathologists who were blinded to flow cytometric analysis results. The criteria for MDS diagnosis were 1) the presence of ≥10% dysplastic cells in any hematopoietic lineage, 2) the exclusion of acute myeloid leukemia (defined by the presence of ≥20% peripheral blood or bone marrow blasts), and 3) the exclusion of reactive etiologies of cytopenia and dysplasia. The criteria for chronic myelomonocytic leukemia (CMML) diagnosis were 1) the presence of persistent peripheral blood monocytosis $\geq 1 \times 10^9$/L and 2) monocytes accounting for more than 10% of the white blood cell differential count [25]. Idiopathic cytopenia of uncertain significance (ICUS) was defined by unexplained mild cytopenia for 6 months of follow-up not fulfilling MDS criteria [26]. Bone marrow cytogenetics, molecular profiling, and flow cytometric scoring for patients with confirmed suspicions of myelodysplastic syndromes were reported elsewhere [14].

## Statistical analysis

Baseline patient characteristics were reported as numbers and percentages for categorical variables and median along with 25[th] and 75[th] percentiles or range for continuous variables. We graphically appraised the agreement in continuous intra-individual RCV between simplified and original gating strategies by examining a scatterplot of differences versus the means of the two variables with the limit of agreement superimposed [27]. We checked for the absence of bias by performing regression analysis of the differences as a function of the means. Intra-class correlation coefficient was used to quantify absolute agreement in continuous intra-individual RCV between simplified and original gating strategies performed with the same peripheral blood sample [28]. We also estimated Cohen's Kappa coefficient to quantify agreement in binary intra-individual RCV with a prespecified threshold of 30.0%. This threshold was used because an intra-individual RCV value for neutrophil MPO expression lower than 30.0% accurately ruled out MDS, with both sensitivity and negative predictive value point estimates of 100%, in a previous study [13].

We assessed comparative accuracy of intra-individual continuous RCV obtained with simplified and original gating strategies by computing and comparing the areas under receiver operating characteristic (ROC) curves [29, 30]. We also reported sensitivity, specificity, positive predictive value, negative predictive value, and likelihood ratio point estimates along with

95% confidence interval (CI) for binary intra-individual RCV with a prespecified threshold of 30.0%.

Two-tailed *P*-values less than 0.05 were considered statistically significant. Analyses were performed using Stata Special Edition version 16.0 (Stata Corporation, College Station, TX, USA).

## Ethics statement

An institutional review board (Comité de Protection des Personnes Sud Méditerranée I, Marseille, France) reviewed and approved the study protocol and the information form, prior to study initiation. According to French regulations, the consent to participate was sought under a regime of "non-opposition" (opt-out): after appropriate written information was delivered, data were collected except in case of written opposition from the patient.

## Results

The analytical sample consisted of 62 consecutive patients with cytomorphological evaluation of bone marrow aspirate confirming MDS in 23 (37%), unconfirming MDS in 32 (52%, including three patients [4.8%] with ICUS), and being uninterpretable in seven (11%). The median age for all participants was 72 years (25–75th percentiles, 64–82) and 27 (43%) were female (Table 1).

Median intra-individual RCV for peripheral blood neutrophil MPO expression obtained with simplified and original gating strategies were 30.7% (25–75th percentiles, 28.6–35.3) and 30.6% (25–75th percentiles, 28.5–34.8), respectively. High level of agreement in continuous intra-individual RCV between the two gating strategies was observed graphically, with mean difference of 0.10 percentage points and 95% limits of agreement ranging from -0.18 to 0.38

**Table 1. Baseline characteristics for consecutive patients with suspicion of myelodysplastic syndromes (n = 62).**

| Characteristics[a] | All patients | | Suspicion of MDS | | | | Uninterpretable BMA | |
| | | | Unconfirmed | | Confirmed[b] | | | |
|---|---|---|---|---|---|---|---|---|
| No. | 62 | (. . .) | 32 | (. . .) | 23 | (. . .) | 7 | (. . .) |
| Female gender, *n (%)* | 27 | (43) | 12 | (37) | 11 | (48) | 4 | (57) |
| Age, median (IQR), *y* | 72 | (64–82) | 71 | (64–81) | 73 | (64–82) | 74 | (65–88) |
| Hemoglobin, median (IQR), *g/dL* | 9.7 | (8.7–11.9) | 9.6 | (8.7–11.6) | 9.4 | (8.6–11.9) | 11.8 | (10.2–13.5) |
| Platelet, median (IQR), ×$10^9$/L | 146 | (63–232) | 142 | (65–258) | 145 | (62–188) | 151 | (63–247) |
| ANC, median (IQR), ×$10^9$/L | 2.3 | (1.4–5.1) | 3.8 | (1.9–6.0) | 1.4 | (1.0–3.2) | 1.4 | (1.0–3.5) |
| Lymphocytes, median (IQR), ×$10^9$/L | 1.0 | (0.8–1.4) | 1.0 | (0.8–1.2) | 1.1 | (0.7–1.5) | 1.2 | (1.0–1.4) |
| Monocytes, median (IQR), ×$10^9$/L | 0.5 | (0.2–0.6) | 0.6 | (0.3–0.7) | 0.2 | (0.1–0.5) | 0.3 | (0.2–0.6) |
| Creatinine, median (IQR), *μmol/L* | 87 | (64–109) | 90 | (71–115) | 79 | (59–95) | 98 | (70–122) |
| C-reactive protein, median (IQR), *mg/L* | 12 | (5–38) | 13 | (5–38) | 8 | (4–36) | 0 | (0–45) |
| ICUS, *n (%)* | 3 | (4.8) | 3 | (9.4) | . . . | (. . .) | . . . | (. . .) |

Abbreviations: ANC = absolute neutrophil count; BMA = bone marrow aspirate; ICUS = idiopathic cytopenia of undetermined significance; IQR = interquartile range (25–75th percentiles); MDS = myelodysplastic syndrome; MPO = myeloperoxidase; RCV = robust coefficient of variation.

[a] Values were missing for absolute neutrophil count (*n* = 2), monocytes (*n* = 2), C-reactive protein (*n* = 20), and creatinine (*n* = 11) concentrations.

[b] Confirmed suspicions of MDS by bone marrow cytomorphology included MDS with excess blasts 1 (n = 9), MDS with multilineage dysplasia (n = 5), MDS with excess blasts 2 (n = 2), MDS with single lineage dysplasia (n = 2), chronic myelomonocytic leukemia (n = 2), MDS with ring sideroblasts (n = 1), and unclassifiable MDS (n = 2).

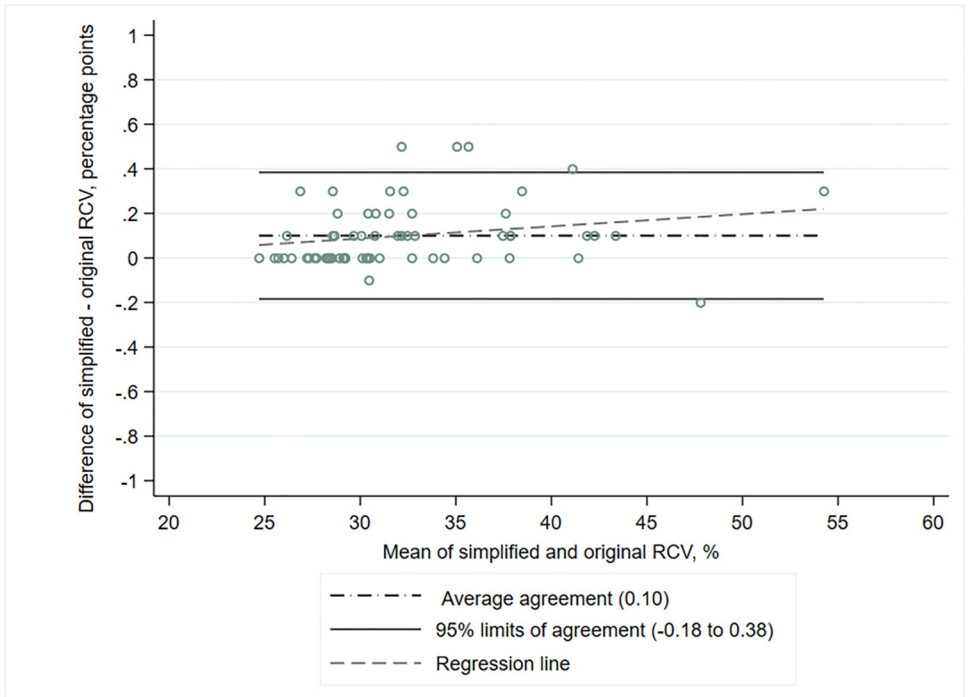

**Fig 3. Agreement in continuous intra-individual robust coefficient of variation between simplified and original flow cytometric gating strategies for quantifying peripheral blood neutrophil myeloperoxidase expression (n = 62).** Abbreviations: RCV = robust coefficient of variation.

(Fig 3). No obvious relationship was found between the difference and the mean (β regression coefficient for slope = 0.005, 95% CI, -0.001 to 0.012, $P$ = .09).

The intra-class correlation coefficient quantifying the absolute agreement in continuous intra-individual RCV between the simplified and original gating strategies was 1.00 (95% CI, 0.99 to 1.00). Perfect agreement in binary intra-individual RCV with a 30.0% prespecified threshold was observed between the two gating strategies (Cohen's Kappa coefficient = 1.00, 95% CI, 0.99 to 1.00, Table 2).

The median intra-individual RCV values for patients with confirmed and unconfirmed suspicion of MDS were 37.4% (range, 30.7–54.1) and 29.2% (range, 24.7–37.8) with the original gating strategy compared with 37.5% (range, 30.8–54.4) and 29.2% (range, 24.7–37.9) with the simplified gating strategy (Table 3). The area under the ROC curve estimates for continuous intra-individual RCV were 0.92 (95% CI, 0.82–0.98) and 0.93 (95% CI, 0.82–0.98) with the original and simplified gating strategies, respectively ($P$ = .32). Using original or simplified

**Table 2. Agreement of binary intra-individual robust coefficient of variation for peripheral blood neutrophil myeloperoxidase expression between simplified and original flow cytometric gating strategies (Cohen's Kappa coefficient = 1.00).**

| Original gating strategy | Simplified gating strategy | | |
|---|---|---|---|
| | RCV < 30% | RCV ≥ 30% | Total |
| RCV < 30% | 24 | 0 | 24 |
| RCV ≥ 30% | 0 | 38 | 38 |
| Total | 24 | 38 | 62 |

Abbreviations: RCV = robust coefficient of variation.

**Table 3. Comparative diagnostic accuracy of intra-individual robust coefficient of variation for peripheral blood neutrophil myeloperoxidase expression between simplified and original flow cytometric gating strategies (n = 55)[a].**

| | Flow cytometric gating strategy | | | |
| --- | --- | --- | --- | --- |
| | Original | | Simplified | |
| Intra-individual RCV, %, median (range) | | | | |
| Confirmed suspicion of MDS (n = 23) | 37.4 | (30.7–54.1) | 37.5 | (30.8–54.4) |
| Unconfirmed suspicion of MDS (n = 32) | 29.2 | (24.7–37.8) | 29.2 | (24.7–37.9) |
| Area under ROC curve (95% CI) | 0.92 | (0.82–0.98) | 0.93 | (0.82–0.98) |
| RCV ≥ 30% | | | | |
| True positive, *n* | 23 | . . . | 23 | . . . |
| False positive, *n* | 13 | . . . | 13 | . . . |
| False negative, *n* | 0 | . . . | 0 | . . . |
| True negative, *n* | 19 | . . . | 19 | . . . |
| Sensitivity, % (95%CI) | 100 | (85–100) | 100 | (85–100) |
| Specificity, % (95%CI) | 59 | (41–76) | 59 | (41–76) |
| PPV, % (95%CI) | 64 | (46–79) | 64 | (46–79) |
| NPV, % (95%CI) | 100 | (82–100) | 100 | (82–100) |

Abbreviations: CI = confidence interval; MDS = myelodysplastic syndrome; NPV = negative predictive value; PPV = positive predictive value; RCV = robust coefficient of variation; ROC = receiver operating characteristic.

[a] The analytical sample consisted of 23 and 32 patients with confirmed and unconfirmed suspicions of myelodysplastic syndrome, after excluding seven patients with uninterpretable bone marrow cytomorphology at baseline.

gating strategy, an intra-individual RCV value <30.0% ruled out MDS for 35% (19 of 55) patients referred for suspected disease, with both sensitivity and negative predictive value point estimates of 100% (Table 3).

After excluding two CMML cases from the study sample, the results remained rather unchanged regarding agreement and comparative accuracy between the original and simplified gating strategies (S3 and S4 Tables). Agreement was high (S5 Table) and diagnostic accuracy estimates were comparable (S6 Table) although less precise due to the relatively limited effective sample size for the two gating strategies, in the validation sample comprising patients with milder level of peripheral blood cytopenia.

## Discussion

This study demonstrates that the simplification of the original gating strategy for quantifying peripheral blood neutrophil MPO expression does not deteriorate the discriminative performance attributes of intra-individual RCV for the diagnosis of MDS. Perfect agreement in intra-individual RCV value and similar diagnostic accuracy for MDS were found between the simplified and the original gating strategies. Using the simplified gating strategy, intra-individual RCV values < 30% accurately ruled out MDS and might obviate the need for bone marrow aspirate in up to 35% of patients referred for suspected disease.

The simplification of the original Boolean gating strategy was not followed by improvement in diagnostic accuracy estimates. Actually, the discriminative accuracy was already high for the original gating strategy, with an area under ROC curve of 0.92, and the potential for improvement was therefore limited.

The simplified gating strategy has the potential to accelerate the quantification of peripheral blood neutrophil MPO expression in routine and to reduce resource utilization although this study was not designed for this purpose [31]. The simplification of the gating strategy may also allow automation and standardization across laboratories. First, the use of premade stable

standardized reagent panels may address the issues of reliability and efficiency inherent to laboratory-developed tests. Lyophilisation has been used to stabilize premixed multicolor reagent cocktails within flow cytometry tubes (Lyotube), for various clinical applications [32]. Lyophilised reagent cocktails provide a simplified way of handling complex multicolor flow cytometry assays, with performance comparable to reference liquid cocktails [32]. Second, computational assistance for visualisation and analysis may reduce analytical time and variability inherent to manual gating [33, 34]. Third, between-laboratory reproducibility as well as inter- and intra-assay precision may be monitored according to current guidelines [35, 36].

In contrast to other assays based on peripheral blood samples [8, 9, 37–40], intra-individual RCV of neutrophil MPO expression has sufficient sensitivity and negative predictive value to safely rule out MDS on its own. Yet, the rate and extent of adoption of flow cytometric analysis for quantifying peripheral blood neutrophil MPO expression in patients with suspected MDS will be determined by issues of cost-effectiveness. Indeed, it is not clear at what point the risks and discomfort of bone marrow aspiration outweight the consequences of misclassification using non-invasive peripheral blood diagnostic test. However, even if the harms are relatively limited, clinicians and patients prefer to avoid bone marrow aspiration if possible. Although it is premature to advocate routine use of this biomarker, it is already an attractive option for patients who are at increased risk for complications of bone marrow aspiration, those who refuse bone marrow aspiration, and those in whom cytomorphological evaluation is uninterpretable.

Our study has several strengths. First, adequate diagnostic reference of MDS was used, with experienced hematologists performing cytomorphological evaluation of bone marrow blinded to index tests. Second, the potential for spectrum bias was minimized by enrolling unselected consecutive patients referred for suspected MDS and using broad inclusion criteria [41]. Third, a prespecified threshold for intra-individual RCV was used to prevent optimistic diagnostic accuracy estimates [41].

The limitations of our study should be acknowledged. First, our study was carried out at a single hospital laboratory and our findings may not apply to other settings. Although the original gating strategy showed satisfactory reproducibility estimates across operators, instrument setup procedures, and laboratories [13], further investigation of performance attributes for the simplified gating strategy is warranted as part of a prospective validation study. Second, cytomorphological evaluation of bone marrow aspirate was uninterpretable at baseline for seven patients for various reasons including dry tap (n = 5), bloody tap (n = 1), and altered aspirate (n = 1). Because the reference diagnosis was not available for these study participants, we cannot exclude that some cases of myelodysplastic syndromes were missed. However, this was unlikely to alter agreement and comparative accuracy estimates between the original and simplified gating strategies. Third, the original gating strategy was somewhat subjective and operators with varying levels of expertise might introduce bias in estimating RCV values. Reassuringly, the RCV value was rather unchanged after excluding granulocytes with lower levels of MPO expression.

## Conclusion

The simplified flow cytometric gating strategy performed as well as the original one for quantifying peripheral blood neutrophil MPO expression, in this study. The two gating strategies can be interchangeably used for ruling out MDS, although the simplified one has the potential to save time and reduce resource utilization in busy flow cytometry laboratories. Yet, prospective validation of the simplified gating strategy would provide additional evidence to support its adoption.

## Supporting information

**S1 Table. Technical information for reagents used for immunocytochemical staining.**
(DOCX)

**S2 Table. Overview of lasers for FACSCanto-II ™ flow cytometer.**
(DOCX)

**S3 Table. Agreement of binary intra-individual robust coefficient of variation for peripheral blood neutrophil myeloperoxidase expression between simplified and original flow cytometric gating strategies after excluding two chronic myelomonocytic leukemia cases from the study sample.**
(DOCX)

**S4 Table. Comparative diagnostic accuracy of intra-individual robust coefficient of variation for peripheral blood neutrophil myeloperoxidase expression between simplified and original flow cytometric gating strategies after excluding two chronic myelomonocytic leukemia cases from the study sample (n = 53).**
(DOCX)

**S5 Table. Agreement of binary intra-individual robust coefficient of variation for peripheral blood neutrophil myeloperoxidase expression between simplified and original flow cytometric gating strategies in external validation sample comprising patients with milder level of peripheral blood cytopenia.**
(DOCX)

**S6 Table. Comparative diagnostic accuracy of intra-individual robust coefficient of variation for peripheral blood neutrophil myeloperoxidase expression between simplified and original flow cytometric gating strategies in external validation sample comprising patients with milder level of peripheral blood cytopenia (n = 23).**
(DOCX)

**S1 File. Deidentified individual participant data.**
(XLS)

## Acknowledgments

The authors thank Drs. Claude Eric Bulabois, Clara Mariette, Martin Carré, Stéphane Courby, Brigitte Pégourié, Anne Thiebaut-Bertrand, and Rémy Gressin for patient recruitment and Drs. Christine Lefebvre and Sylvie Tondeur, for cytogenetic and molecular analysis. The authors are indebted to Séverine Beatrix, Laure Dusset, Ghislaine Del-Vecchio, Richard Di Schiena, Michel Drouin, Claire Gasquez, Frédérique Martinez, Karine Nicolino and Christine Vallet for their technical assistance.

## Author Contributions

**Conceptualization:** Tatiana Raskovalova, Marie-Christine Jacob, José Labarère.

**Data curation:** Laura Scheffen.

**Formal analysis:** Tatiana Raskovalova, Marie-Christine Jacob, Claire Vettier, Bénédicte Bulabois, Gautier Szymanski, Simon Chevalier, Sophie Park, José Labarère.

**Funding acquisition:** Tatiana Raskovalova.

**Investigation:** Tatiana Raskovalova.

**Project administration:** Nicolas Gonnet.

**Supervision:** Tatiana Raskovalova.

**Writing – original draft:** Tatiana Raskovalova, Marie-Christine Jacob, José Labarère.

**Writing – review & editing:** Laura Scheffen, Claire Vettier, Bénédicte Bulabois, Gautier Szymanski, Simon Chevalier, Nicolas Gonnet, Sophie Park.

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
