## [Decision Letter · Decision Letter 0]

17 May 2022

PONE-D-22-04816Comparative diagnostic accuracy between simplified and original flow cytometric gating strategies for peripheral blood neutrophil myeloperoxidase expression in ruling out myelodysplastic syndromes.PLOS ONE

Dear Dr. Labarere,

Thank you for submitting your manuscript to PLOS ONE. After careful consideration, we feel that it has merit but does not fully meet PLOS ONE’s publication criteria as it currently stands. Therefore, we invite you to submit a revised version of the manuscript that addresses the points raised during the review process. The reviewers were interested in the study and the proposed approach to the analysis. However, they felt the manuscript would benefit from additional information, particularly in relation to the methodological nature of this study. I would be grateful if you could consider their detailed comments provided below.

We look forward to receiving your revised manuscript.

Kind regards,

Eduard Shantsila

Academic Editor

PLOS ONE

Journal Requirements:

“Becton Dickinson Bioscience provided antibodies free of charge (Dr Raskovalova).

Statistical analysis was performed within the Grenoble Alpes Data Institute (ANR-15-IDEX-02). (José Labarère)

This research received no other specific grant from any funding agency in the public, commercial, or not-for-profit sectors.”

“Becton Dickinson Biosciences provided antibodies free of charge. Statistical analysis was 337 performed within the Grenoble Alpes Data Institute (ANR-15-IDEX-02). This research 338 received no other specific grant from any funding agency in the public, commercial, or not339 for-profit sectors.”

“Becton Dickinson Bioscience provided antibodies free of charge (Dr Raskovalova).

Statistical analysis was performed within the Grenoble Alpes Data Institute (ANR-15-IDEX-02). (José Labarère)

This research received no other specific grant from any funding agency in the public, commercial, or not-for-profit sectors.”

Reviewers' comments:

Reviewer's Responses to Questions

**Comments to the Author**

1. Is the manuscript technically sound, and do the data support the conclusions?

Reviewer #1: Partly

Reviewer #2: Partly

2. Has the statistical analysis been performed appropriately and rigorously? 

Reviewer #1: Yes

Reviewer #2: Yes

3. Have the authors made all data underlying the findings in their manuscript fully available?

Reviewer #1: Yes

Reviewer #2: No

4. Is the manuscript presented in an intelligible fashion and written in standard English?

Reviewer #1: Yes

Reviewer #2: Yes

5. Review Comments to the Author

Reviewer #1: General remarks

This manuscript describes the methodological component of a very promising method to reduce the invasive bone marrow procurements among patients that are suspected of MDS. By means of a simple yet accurate method, 35% of patients may be ruled out for MDS and benefits of using this technique are clearly described. The gating strategy of diagnostic flow cytometric assays are an important component of performing the assay and should indeed be thoroughly investigated to ensure accuracy as well as reproducibility of the method in clinical practice. I have therefore read this manuscript and the related manuscripts with great interest and enjoyment. Nevertheless, I have a number of major and minor remarks that could further improve the manuscript. Please carefully address the different comments.

Comments

Major

P5 Line 63 (Introduction): I surely agree that patients are exposed to unnecessary bone marrow aspiration-related discomforts and harms. I would recommend to elaborate in perhaps a single/2 sentences about these, as this helps the reader to understand the necessity of simplification of the diagnostic process.

P5 Line 69 (Introduction): What made the gating strategy so complex? Please elaborate shortly on this point.

P6 Line 91 (Methods): What is the rationale for choosing the cut-off of 50 years and older for inclusion?

P7 Line 114-115 and Figure 1: Why did you leave out the cells low in FSC-Peak Height and FSC-Area? These cells may just as well be singlets.

(please see reference: Theresia M. Westers et al. Immunophenotypic analysis of erythroid dysplasia in myelodysplastic syndromes. A report from the IMDSFlow working group. Haematologica 2017; https://doi.org/10.3324/haematol.2016.147835).

P7 Line 121 (Methods): As this is a more methodic paper, I would recommend to elaborate thoroughly on how you set the gating threshold for MPO positivity. My main question would be when repeating this strategy is if I should include the red (granulocytes) and green (monocytes) cells that are MPOdim as the purple population seems truly negative. As you work with rCV as diagnostic criterion these cells may influence the reported rCV values. Please include an additional plot with a clear MPO negative granulocyte and monocyte population.

P7 Line 121 (Methods) and Figure 1: Did you also compare the use of a rectangular gate compared to use of a polygonal gate for MPO positivity?

(for example on this please see: Stefan G.C. Mestrum, et al. Optimized gating strategy and supporting flow cytometry data for the determination of the Ki-67 proliferation index in the diagnosis of myelodysplastic syndrome, Data in Brief 2022; https://doi.org/10.1016/j.dib.2022.107976).

The use of rectangular gates is generally more straightforward to implement and easier to learn for operators in busy clinical centers.

P8 Line 133 (Methods): What is the rationale for including gating of the monocytes, lymphocytes and eosinophils in the full gating strategy, but not in the simplified gating strategy? Please elaborate on this or correct the gating the strategy.

P11 Line 205 (Results): Please exclude the CMML cases as these are part of the MDS/MPN overlapping syndrome. MDS/MPN is considered as a separate disease category as compared to MDS according to the WHO classification.

P15 Line 271 (Discussion): Definition of the gating threshold is elaborated on here. I recommend to clearly state how you defined this gating threshold in the Methods section, as this is the most important component of your analysis strategy. Comparison of the reproducibility of the different gating protocols can be left in the Discussion.

P16 Line 294 (Discussion): Automatization and standardization are both important aspects of this interesting diagnostic method. It would be interesting to explain more about these aspects from a methodological perspective in the Discussion section. Could this for example be analyzed with techniques like FlowSOM? What are your recommendations for further standardization (e.g. antibody panel designs and quality ensurance of gating procedures among clinical centers)?

P17 Line 319 (Discussion): Why were these cytomorphological evaluations uninterpretable for these 7 patients? Is this a real limitation of the study considering this reason?

Minor

P3 Line 29 (Abstract): I believe that “flow cytometric analysis” instead of “flow cytometry analysis” is the correct term. Please adjust this accordingly.

P6 Line 95 (Methods): The subtitle “Index test” is in my opinion not explanatory for the content included. Please adjust the title to be more clear.

Furthermore, this paragraph only includes 1 sentence, please allocate it to another paragraph or elaborate more on its content.

I would recommend to state that the operators were blinded to the definitive diagnosis of the participants, as other techniques besides cytomorphological evaluation may have given the operators information about the definitive diagnosis.

P7 Line 102 (Methods): Please combine the paragraphs “Blood staining” and “Fixation and permeabilization”. A possible title for the combined paragraph could be “Immunocytochemical staining”

P7 Line 107 (Methods): Please use the full name for MPO first and abbreviate between brackets. Later on in section “Original gating strategy” (P7 Line 121) you use the full name again.

P7 Line 114 and P8 Line 130 (Methods): I think the term viable is not correct here. I think that only debris and doublets are excluded by the use of the FSC and SSC characteristics. Viability assessment should be performed with dyes that detect this accordingly. Please adjust the term viable to exclusion of debris and doublets.

P7 Line 119 (Methods): The description of the phenotype that is gated by Boolean intersection could be improved. The AND NOT statements make it hard for the reader to understand what the actual phenotype of the cells is. More straightforward would be for example: [CD15+ CD14-][CD45low CD16 high] etc. Please describe this thoroughly.

P8 Line 128 (Methods): Please correct to “In the simplified gating strategy”

P8 Line 129 (Methods): Histograms are not shown in Fig. 2, while these are named in the text. Please explain what these histograms refer to or leave histograms out of the text.

P8-P9 Lines 149 and 152 (Methods): Please be consistent in the annotation of CD markers with fluorochromes. Now you used CD64-FITC (with stripe) first and then CD64 FITC (without a stripe).

P10 Line 192 (Results): What are the demographics of the aspirates unconfirming MDS? I could not find information on this in the referred paper as well as this one. A suggestion would be to separately report the characteristics of patients with confirmed suspicion of MDS and unconfirmed suspicion of MDS in Table 1.

Discussion section: Please restructure the Discussion section here and there to avoid redundancies. For example, the paragraph at line 271-281 and the paragraph at line 291-294 seem to contain redundancies.

Reviewer #2: In this manuscript, Raskovalova et al. have investigated diagnostic accuracy of a simple gating strategy for neutrophil myeloperoxidase (MPO) assessment in peripheral blood samples. Despite the interest and novelty of this work, several issues should be addressed.

This manuscript is intended to be a method article and to propose a standardized procedure for peripheral blood cell immunophenotyping for differential diagnosis of myelodysplastic syndromes (MDS). All technical steps should be better explained to give to the readers all the information needed to perform this procedure in clinical practice in a standardized manner. Therefore, better explain this “index test”, antibodies (with fluorochromes and clones) used, lasers equipped on the FACSCanto-II cytometer, controls used for PMT voltage setting, quality controls of the instrument, and compensation.

The Authors should better discuss the choice of used antigens for immunophenotyping (CD15, CD14, CD16, CD11b, and CD64), and should indicate literature for international guidelines on neutrophil phenotyping used. Not clear why the Authors are trying to remove CD64 from the analysis. Just a compensation issue? Have you tried other fluorochrome combinations? CD64 is important in neutrophil phenotype as also described in some OMIP articles (e.g., Zhu G, Brayer J, Padron E, Mulé JJ, Mailloux AW. OMIP-049: Analysis of Human Myelopoiesis and Myeloid Neoplasms. Cytometry A. 2018 Oct;93(10):982-986).

Cytogenetics and FISH analysis are essential for MDS diagnosis and classification. This information should be added. Moreover, if possible, IPSS should be included too.

The Authors are showing that there is concordance between the “original” vs “novel” gating strategy; however, despite this result could be seen as a positive result showing the non-inferiority of this method in identifying MPO+ neutrophils, this procedure seems not to improve clinical definition of MDS and seems not have a clinical impact for differential diagnosis of MDS, ICUS, and normal conditions. Please better discuss this point.

Some introductions on MPO in MDS should be added.

FCS files should be deposited on appropriate public databases for methodological papers (e.g., https://flowrepository.org/) without patient-related information.

A validation cohort should be added (even retrospective using “old” FCS files from the same institution).

Abbreviations should be carefully checked and refer to myeloperoxidase as MPO.

6. PLOS authors have the option to publish the peer review history of their article (what does this mean?). If published, this will include your full peer review and any attached files.

Reviewer #1: No

Reviewer #2: **Yes: **Valentina Giudice

---

## [Author Response · Author response to Decision Letter 0]

1 Jul 2022

Dear Prof. Shantsila,

We appreciate your decision to consider a revision of our manuscript entitled “Comparative diagnostic accuracy between simplified and original flow cytometric gating strategies for peripheral blood neutrophil myeloperoxidase expression in ruling out myelodysplastic syndromes” (PONE-D-22-04816).

The funders had no role in study design, data collection and analysis, decision to publish, or preparation of the manuscript

We are very grateful to the reviewers for their thorough examination of our manuscript and for the practical presentation of their comments. Below, please find item-by-item responses to the reviewers’ comments, which are included verbatim. All page and paragraph numbers refer to locations in the revised manuscript.

We hope that the new version of our paper, which has been rewritten taking the reviewers' comments into account, is deemed worthy of publication in PLOS ONE. 

Sincerely,

Tatiana Raskovalova 

Reply: The manuscript has been revised in compliance with PLOS ONE’s style requirements.

2. Please provide additional details regarding participant consent. In the ethics statement in the Methods and online submission information, please ensure that you have specified what type you obtained (for instance, written or verbal, and if verbal, how it was documented and witnessed). If your study included minors, state whether you obtained consent from parents or guardians. If the need for consent was waived by the ethics committee, please include this information. Once you have amended this/these statement(s) in the Methods section of the manuscript, please add the same text to the “Ethics Statement” field of the submission form (via “Edit Submission”).

Reply: Additional details regarding participant consent have been included in the Ethics statement as part of the Methods section. The Ethics Statement now reads as follows:

“An institutional review board (Comité de Protection des Personnes Sud Méditerranée I, Marseille, France) reviewed and approved the study protocol and the information form, prior to study initiation. According to French regulations, the consent to participate was sought under a regime of “non-opposition” (opt-out): after appropriate written information was delivered, data were collected except in case of written opposition from the patient.” (page 11)

The Ethics Statement field of the online submission form has been edited accordingly.

3. Please include the Role of Funder statement in your cover letter; we will change the online submission form on your behalf.

The Role of Funder statement should read as follows:

We thank the Editorial Assistant for updating the online submission form accordingly.

4. Please note that funding information should not appear in the Acknowledgments section or other areas of your manuscript. We will only publish funding information present in the Funding Statement section of the online submission form. Please remove any funding-related text from the manuscript and let us know how you would like to update your Funding Statement. 

As requested, the funding information has been removed from the manuscript. The Funding Statement field of the online submission form should read as follows:

 “Becton Dickinson Bioscience provided antibodies free of charge (Dr Raskovalova).

Statistical analysis was performed within the Grenoble Alpes Data Institute (ANR-15-IDEX-02). (José Labarère).

This research received no other specific grant from any funding agency in the public, commercial, or not-for-profit sectors.”

We thank the Editorial Assistant for updating the online submission form accordingly.

Reviewer #1

1. P5 Line 63 (Introduction): I surely agree that patients are exposed to unnecessary bone marrow aspiration-related discomforts and harms. I would recommend to elaborate in perhaps a single/2 sentences about these, as this helps the reader to understand the necessity of simplification of the diagnostic process.

As suggested by this Reviewer, we elaborated on this important issue. Changes to the manuscript are highlighted below:

“Bone marrow aspiration is an invasive procedure, with up to 20% of patients reporting a moderate level of pain for 7 days or more. Although infrequent, procedure-related complications (hemorrhage and infection) may be associated with significant morbidity or even be life-threatening.”

2. P5 Line 69 (Introduction): What made the gating strategy so complex? Please elaborate shortly on this point.

To address this Reviewer’s comment, we added a sentence explaining what made the original gating strategy complex: 

“The Boolean combination of gates was time consuming and relied on operator’s expertise for individualizing cell subpopulations (neutrophils, eosinophils, lymphocytes, and monocytes) before measuring neutrophil MPO expression.”

3. P6 Line 91 (Methods): What is the rationale for choosing the cut-off of 50 years and older for inclusion?

We appreciate the opportunity to clarify this point. The incidence of myelodysplastic syndromes increases with advancing age, with incidence rates ranging from 0.1 cases per 100,000 person-year for those younger than 40 years to 56 cases per 100,000 person-year at an age of 80 years or older. Myelodysplastic syndromes is predominantly a diagnosis of older adults, with a median age at diagnosis of 77 years. Given the very low prevalence of myelodysplastic syndromes below the age of 50 years (See Zeidan AM et al. Blood Rv 2019;34:1-15), enrolling younger participants would have produced imprecise and unreliable diagnostic accuracy estimates for this subgroup of patients.”

We have modified the Methods Section in response to this comment as follows:

“Given the very low prevalence of myelodysplastic syndromes below the age of 50 years, enrolling younger participants would have produced imprecise estimates for this subgroup of patients.”

4. P7 Line 114-115 and Figure 1: Why did you leave out the cells low in FSC-Peak Height and FSC-Area? These cells may just as well be singlets (please see reference: Theresia M. Westers et al. Immunophenotypic analysis of erythroid dysplasia in myelodysplastic syndromes. A report from the IMDSFlow working group. Haematologica 2017.

We thank this Reviewer for the suggested reference and we have modified Figure 1 panel A accordingly.

5. P7 Line 121 (Methods): As this is a more methodic paper, I would recommend to elaborate thoroughly on how you set the gating threshold for MPO positivity. My main question would be when repeating this strategy is if I should include the red (granulocytes) and green (monocytes) cells that are MPOdim as the purple population seems truly negative. As you work with rCV as diagnostic criterion these cells may influence the reported rCV values. Please include an additional plot with a clear MPO negative granulocyte and monocyte population.

We appreciate the opportunity to clarify this important issue. Actually, there is no gating threshold for MPO positivity. In the original Figure 1 Panel G, the MPO gate was not conditioned on the events that were visualized (i.e., neutrophils, monocytes and lymphocytes) but instead was based on the following Boolean approach: [CD15+ CD14-] [CD16+ CD11b+] with exclusion of [CD45high CD16 low] [CD14+ CD15dim/-] [CD15- CD14-]. 

To address this Reviewer’s concern, we modified Figure 1 Panel G in the revised manuscript. The MPO gate now appears to be conditioned on granulocytes only after excluding eosinophils, monocytes and lymphocytes with Boolean gating. Robust coefficient of variation (RCV) estimate for MPO was based on all visualised granulocytes (red dots) without threshold whereas monocytes (green dots) and lymphocytes (purple dots) do not appear anymore in Figure 1 Panel G.

6. P7 Line 121 (Methods) and Figure 1: Did you also compare the use of a rectangular gate compared to use of a polygonal gate for MPO positivity? (for example on this please see: Stefan G.C. Mestrum, et al. Optimized gating strategy and supporting flow cytometry data for the determination of the Ki-67 proliferation index in the diagnosis of myelodysplastic syndrome, Data in Brief 2022; https://doi.org/10.1016/j.dib.2022.107976). The use of rectangular gates is generally more straightforward to implement and easier to learn for operators in busy clinical centers.

We have included the reference suggested by this Reviewer as follows:

“Although rectangular gating is more straightforward to implement and easier to learn for operators, we used manually placed polygonal gates as described in a previous study.”

7. P8 Line 133 (Methods): What is the rationale for including gating of the monocytes, lymphocytes and eosinophils in the full gating strategy, but not in the simplified gating strategy? Please elaborate on this or correct the gating the strategy.

We appreciate the opportunity to clarify this point. Our original gating strategy was based on Boolean gating using “AND”, “OR”, and “NOT” logic, in order to individualize cell subpopulations (negative selection of granulocytes). In contrast, the simplified gating strategy was hierarchical (positive selection of granulocytes), obviating Boolean operators and reducing the number of gates from 12 to 4 and the number of dot plots from 7 to 4. 

We have modified the Methods Section in response to this comment as follows:

“The original gating strategy was based on Boolean combination of gates, using “AND”, “OR”, and “NOT” logic, in order to individualize cell subpopulations (negative selection of granulocytes).”

“The simplified gating strategy was hierarchical (positive selection of granulocytes) (Fig 2), obviating Boolean operators and reducing the number of gates from 12 to 4 and the number of dot plots from 7 to 4.”

8. P11 Line 205 (Results): Please exclude the CMML cases as these are part of the MDS/MPN overlapping syndrome. MDS/MPN is considered as a separate disease category as compared to MDS according to the WHO classification.

To address this Reviewer’s concern, we repeated the analysis after excluding the two CMML cases. The results were rather unchanged regarding agreement and comparative accuracy between the original and simplified gating strategies. Changes in the revised manuscript in response to the above are highlighted below:

“After excluding two CMML cases from the study sample, the results remained rather unchanged for agreement and comparative accuracy between the original and simplified gating strategies (Tables S3–S4).”

9. P15 Line 271 (Discussion): Definition of the gating threshold is elaborated on here. I recommend to clearly state how you defined this gating threshold in the Methods section, as this is the most important component of your analysis strategy. Comparison of the reproducibility of the different gating protocols can be left in the Discussion.

This comment is similar to comment#5 from this Reviewer. Actually, there was no gating threshold for MPO positivity. Please see our response to that comment.

10. P16 Line 294 (Discussion): Automatization and standardization are both important aspects of this interesting diagnostic method. It would be interesting to explain more about these aspects from a methodological perspective in the Discussion section. Could this for example be analyzed with techniques like FlowSOM? What are your recommendations for further standardization (e.g. antibody panel designs and quality ensurance of gating procedures among clinical centers)?

We appreciate this comment and have expanded the discussion on the potential for automation and standardization of this diagnostic test as follows:

“There are various ways for improving standardization and automation. First, the use of premade stable standardized reagent panels may address the issues of reliability and efficiency inherent to laboratory-developed tests. Lyophilisation is a method which has been used to stabilize premixed multicolor reagent cocktails within flow cytometry tubes (Lyotube), for various clinical applications. Lyophilised reagent cocktails provide a simplified way of handling complex multicolor flow cytometry assays, with performance comparable to reference liquid cocktails. Second, computational assistance for visualisation and analysis may reduce analytical time and variability inherent to manual gating. Third, between-laboratory reproducibility as well as inter- and intra-assay precision may be monitored according to current guidelines.” 

11. P17 Line 319 (Discussion): Why were these cytomorphological evaluations uninterpretable for these 7 patients? Is this a real limitation of the study considering this reason?

We elaborated on this limitation in our revised manuscript as highlighted below:

“Second, cytomorphological evaluation of bone marrow aspirate was uninterpretable at baseline for seven patients for various reasons including dry tap (n = 5), bloody tap (n = 1), and altered aspirate (n = 1). Because the reference diagnosis was not available for these study participants, we cannot exclude that some cases of myelodysplastic syndromes were missed. However, this was unlikely to alter agreement and comparative accuracy estimates between the original and simplified gating strategies.”

Minor

12. P3 Line 29 (Abstract): I believe that “flow cytometric analysis” instead of “flow cytometry analysis” is the correct term. Please adjust this accordingly.

This has been corrected accordingly.

13. P6 Line 95 (Methods): The subtitle “Index test” is in my opinion not explanatory for the content included. Please adjust the title to be more clear. Furthermore, this paragraph only includes 1 sentence, please allocate it to another paragraph or elaborate more on its content. I would recommend to state that the operators were blinded to the definitive diagnosis of the participants, as other techniques besides cytomorphological evaluation may have given the operators information about the definitive diagnosis.

As suggested by this Reviewer the subtitle has been changed to better reflect the content of the section (i.e. “Flow cytometric analysis”). 

14. P6 Line 95 (Methods): Furthermore, this paragraph only includes 1 sentence, please allocate it to another paragraph or elaborate more on its content. I would recommend to state that the operators were blinded to the definitive diagnosis of the participants, as other techniques besides cytomorphological evaluation may have given the operators information about the definitive diagnosis.

This sentence was rephrased accordingly and allocated to another paragraph.

15. P7 Line 102 (Methods): Please combine the paragraphs “Blood staining” and “Fixation and permeabilization”. A possible title for the combined paragraph could be “Immunocytochemical staining”

We thank this Reviewer for his/her suggestion. We have combined the two paragraphs accordingly.

16. P7 Line 107 (Methods): Please use the full name for MPO first and abbreviate between brackets. Later on in section “Original gating strategy” (P7 Line 121) you use the full name again.

In response to this comment, MPO abbreviation has been used throughout the manuscript, after first occurrence.

17. P7 Line 114 and P8 Line 130 (Methods): I think the term viable is not correct here. I think that only debris and doublets are excluded by the use of the FSC and SSC characteristics. Viability assessment should be performed with dyes that detect this accordingly. Please adjust the term viable to exclusion of debris and doublets.

We appreciate this comment and have modified the sentence as suggested by the Reviewer.

18. P7 Line 119 (Methods): The description of the phenotype that is gated by Boolean intersection could be improved. The AND NOT statements make it hard for the reader to understand what the actual phenotype of the cells is. More straightforward would be for example: [CD15+ CD14-][CD45low CD16 high] etc. Please describe this thoroughly.

We appreciate this Reviewer’s suggestion and have rephrased this sentence as follows:

“Mature neutrophils were visualized by [CD15+ CD14-] [CD45low CD16 high] [CD16+ CD11b+] and selected by the following Boolean intersection: [CD15+ CD14-] [CD16+ CD11b+] after excluding [CD45high CD16 low] [CD14+ CD15dim/-] [CD15- CD14-] (panel F).”

19. P8 Line 128 (Methods): Please correct to “In the simplified gating strategy”

We thank this Reviewer for pointing out this typo.

20. P8 Line 129 (Methods): Histograms are not shown in Fig. 2, while these are named in the text. Please explain what these histograms refer to or leave histograms out of the text.

We agree with this Reviewer that histograms are not shown in Fig. 2 and apologize for this mistake. This sentence has been modified accordingly in the revised manuscript.

21. P8-P9 Lines 149 and 152 (Methods): Please be consistent in the annotation of CD markers with fluorochromes. Now you used CD64-FITC (with stripe) first and then CD64 FITC (without a stripe).

We thank this Reviewer for pointing out this inconsistency.

22. P10 Line 192 (Results): What are the demographics of the aspirates unconfirming MDS? I could not find information on this in the referred paper as well as this one. A suggestion would be to separately report the characteristics of patients with confirmed suspicion of MDS and unconfirmed suspicion of MDS in Table 1.

To address this Reviewer’s concern, summary statistics for baseline characteristics stratified according to reference diagnosis have been reported in Table 1, in the revised manuscript. This information was also reported in Table 1 in Raskovalova T., et al. Ann Hematol 2021;100:1149-58.

23. Discussion section: Please restructure the Discussion section here and there to avoid redundancies. For example, the paragraph at line 271-281 and the paragraph at line 291-294 seem to contain redundancies.

We have modified the Discussion section in order to avoid redundancies in the revised manuscript.

Reviewer #2:

1. This manuscript is intended to be a method article and to propose a standardized procedure for peripheral blood cell immunophenotyping for differential diagnosis of myelodysplastic syndromes (MDS). All technical steps should be better explained to give to the readers all the information needed to perform this procedure in clinical practice in a standardized manner. Therefore, better explain this “index test”, antibodies (with fluorochromes and clones) used, lasers equipped on the FACSCanto-II cytometer, controls used for PMT voltage setting, quality controls of the instrument, and compensation.

To address this Reviewer’s concern, we elaborated on technical specifications in the Methods section of the revised manuscript, including:

- antibodies along with fluorochromes and clones (page 7 and Table S1)

- lasers equipped on the FACSCanto-II cytometer (page 9 and Table S2)

- controls used for PMT voltage setting, quality controls of the instrument, and compensation (Page 9) 

Reassuringly, a previous study found satisfactory reproducibility for RCV between laboratories and setup procedures (i.e., manufacturer’s recommendations [cytometer setup and tracking research beads], FranceFlow and EuroFlow instrument setups) across five healthy controls and five MDS cases (See Raskovalova T, et al. Haematologica 2019;104(12):2382-90, Online Supplemental Table 9).

2. The Authors should better discuss the choice of used antigens for immunophenotyping (CD15, CD14, CD16, CD11b, and CD64), and should indicate literature for international guidelines on neutrophil phenotyping used. 

We appreciate the opportunity to clarify this point. In the revised manuscript, we elaborated on the rationale supporting the choice of antigens used for immunophenotyping and included relevant citations, as follows: 

“CD15 is considered a pan-granulocytic phenotypic marker (15-17) while CD14 was used as a monocyte phenotypic marker (15, 17, 18). CD16, a marker expressed by neutrophils, was used to exclude eosinophils (CD16 low). CD11b is a classical human neutrophil phenotypic marker (15-17, 19).” 

3. Not clear why the Authors are trying to remove CD64 from the analysis. Just a compensation issue? Have you tried other fluorochrome combinations? CD64 is important in neutrophil phenotype as also described in some OMIP articles (e.g., Zhu G, Brayer J, Padron E, Mulé JJ, Mailloux AW. OMIP-049: Analysis of Human Myelopoiesis and Myeloid Neoplasms. Cytometry A. 2018 Oct;93(10):982-986).

We apologize for the confusion introduced by the use of CD64 in the manuscript. Actually, we previously reported that CD64 did not contribute to the individualization of neutrophils (See Raskovalova T, et al. Haematologica 2019;104(12):2382-90 and Raskovalova T, et al. Ann Hematol 2021;100:1149-58). Originally, fluorescent intensity of CD64 (CD64low-neutrophils) was used to assess whether mature neutrophils were properly selected by our gating strategy (CD15, CD14, CD16, CD11b). Based on our previous research, we acknowledge that CD64 is not a relevant candidate marker for refining our gating strategy. For clarity purpose, we have removed reference to CD64 from the revised manuscript.

4. Cytogenetics and FISH analysis are essential for MDS diagnosis and classification. This information should be added. Moreover, if possible, IPSS should be included too.

We agree with this Reviewer that cytogenetics and FISH are essential information for the diagnosis and classification of myelodyplastic syndromes. These data have been published extensively in Raskovalova T, et al. Ann Hematol 2021;100:1149-58 (Table 2). To avoid redundancies between the two papers, we did not include this information in the revised manuscript. However, we would be willing to include this information as a supplemental Table if the Editor or the Reviewer believe that it would be critical for the reader. Changes to the manuscript are highlighted below:

“Bone marrow cytogenetics, molecular profiling, and flow cytometric scoring, for patients with confirmed suspicions of myelodysplastic syndromes were reported elsewhere.”

5. The Authors are showing that there is concordance between the “original” vs “novel” gating strategy; however, despite this result could be seen as a positive result showing the non-inferiority of this method in identifying MPO+ neutrophils, this procedure seems not to improve clinical definition of MDS and seems not have a clinical impact for differential diagnosis of MDS, ICUS, and normal conditions. Please better discuss this point.

We agree with this Reviewer that the simplification of the gating strategy might have been followed by improvement in accuracy estimates. Actually, the discriminative accuracy for the original gating strategy was already high, with an area under ROC curve of 0.92, and the potential for improvement was therefore limited. Unsurprisingly, the simplified and original gating strategies yielded comparable diagnostic accuracy. However, the simplified gating strategy has the potential to accelerate the quantification of peripheral blood neutrophil MPO expression in routine and to reduce resource utilization. We have emphasized this point in the Discussion section of the revised manuscript.

“The simplification of the original Boolean gating strategy was not followed by improvement in diagnostic accuracy estimates. Actually, the discriminative accuracy was already high for the original gating strategy, with an area under ROC curve of 0.92, and the potential for improvement was therefore limited.”

6. Some introductions on MPO in MDS should be added.

In response to this Reviewer’s comment, we developed the rationale supporting MPO cytoplasmic expression as a biomarker for myelodysplastic syndromes as follows:

“Myeloperoxidase (MPO) is an enzyme synthetized during myeloid differentiation and that constitutes the major component of neutrophil azurophilic granules (10). MPO cytoplasmic expression is associated with degranulation of mature granulocytes (11), a classical dysplastic feature of MDS (12). MPO cytoplasmic expression can be analyzed using various approaches, including immuno-cytochemical staining and flow cytometry.”

7. FCS files should be deposited on appropriate public databases for methodological papers (e.g., https://flowrepository.org/) without patient-related information.

We agree with this Reviewer that deidentified FCS files should be deposited on a public databasis, for transparency purpose. Yet we regret that the suggested website is temporarily unavailable.

8. A validation cohort should be added (even retrospective using “old” FCS files from the same institution).

As suggested by this Reviewer, we examined agreement and comparative accuracy between the original and simplified gating strategies in a validation sample made of individual participant data with milder level of peripheral blood cytopenia. Diagnostic accuracy estimates were comparable although less precise due to the relatively limited effective sample size for the two gating strategies. 

We incorporated this new information in Methods and Results secton of the revised manuscript as follows:

“Using the individual data obtained from patients with milder level of peripheral blood cytopenia, we assessed agreement and comparative accuracy of RCV for the diagnosis of MDS, between the original and simplified gating strategies.” (Page 6)

“Agreement was high (Table S5) and diagnostic accuracy estimates were comparable (Table S6) although less precise due to the relatively limited effective sample size for the two gating strategies, in the validation sample comprising patients with milder level of peripheral blood cytopenia.” (Page 17)

9. Abbreviations should be carefully checked and refer to myeloperoxidase as MPO.

We have carefully reviewed the manuscript for consistency in abbreviations.

---

## [Decision Letter · Decision Letter 1]

22 Jul 2022

PONE-D-22-04816R1Comparative diagnostic accuracy between simplified and original flow cytometric gating strategies for peripheral blood neutrophil myeloperoxidase expression in ruling out myelodysplastic syndromes.PLOS ONE

Dear Dr. Labarere,

Thank you for submitting your manuscript to PLOS ONE. After careful consideration, we feel that it has merit but does not fully meet PLOS ONE’s publication criteria as it currently stands. Therefore, we invite you to submit a revised version of the manuscript that addresses the points raised during the review process.

We look forward to receiving your revised manuscript.

Kind regards,

Eduard Shantsila

Academic Editor

PLOS ONE

Reviewers' comments:

Reviewer's Responses to Questions

**Comments to the Author**

1. If the authors have adequately addressed your comments raised in a previous round of review and you feel that this manuscript is now acceptable for publication, you may indicate that here to bypass the “Comments to the Author” section, enter your conflict of interest statement in the “Confidential to Editor” section, and submit your "Accept" recommendation.

Reviewer #1: (No Response)

Reviewer #2: All comments have been addressed

2. Is the manuscript technically sound, and do the data support the conclusions?

Reviewer #1: Partly

Reviewer #2: Yes

3. Has the statistical analysis been performed appropriately and rigorously? 

Reviewer #1: Yes

Reviewer #2: Yes

4. Have the authors made all data underlying the findings in their manuscript fully available?

Reviewer #1: Yes

Reviewer #2: No

5. Is the manuscript presented in an intelligible fashion and written in standard English?

Reviewer #1: Yes

Reviewer #2: Yes

6. Review Comments to the Author

Reviewer #1: I would like to thank the authors for taking the time to revise the manuscript and responding to the different comments. I am still of opinion that this is methodological paper contains relevant and important information for executing this promising diagnostic approach for MDS. However, I am still of opinion that the methodology is not fully clear. As this is a methodological paper, I think the clarity of the methodology to the readers is of utmost importance and will be an important asset to facilitate the clinical application of this diagnostic approach.

Major

Figure 1G: Following up on my last comment about the gating threshold in this plot, but now more specifically. What are the gray dots in this plot (I noticed these when I downloaded the Figure and looked very closely)? Are those MPO negative granulocytes and shouldn’t those be included in your analysis as well, as degranulation occurred in these granulocytes? Excluding granulocytes with low MPO expression levels creates a bias in your RCVs (although small in the displayed case). However, if there are more granulocytes with such low MPO expression levels, readers will exclude those based on the method displayed here, which is incorrect and will tremendously influence the resulting RCVs. To address this comment I would like to see the following: Please analyze the RCVs on all granulocytes without excluding any based on MPO expression levels. This will prevent any potential bias in the RCVs of MPO expression in the granulocytes.

Minor

P5 Line 61: Please leave out Indeed in the beginning of the sentence.

P5 Line 68: This part of the sentence is not correctly written: “that constitutes the major component of neutrophil azurophilic granules”.

P5 Line 70: “MPO cytoplasmic expression” should be corrected to “”cytoplasmic expression of MPO”

P5 Line 71: “immuno-cytochemical” should be corrected to “immunocytochemical”

P5 Line 76: Thank you for adding this passage to the manuscript. One small remark about part of this sentence: “relied on operator’s expertise” . Perhaps the authors can correct this to: “relied on the expertise of the operators”.

P6 Line 104: In this part of the sentence “we assessed agreement and comparative accuracy of RCV”, the abbreviation of RCV is first used and should be written in full with abbreviation between brackets. I also miss a word in the sentence. Please correct this part of the sentence to: “we assessed the agreement and comparative accuracy of the RCV. ”

P7 Line 199-128: Please remove the bullet points and write this down in narrative form.

P10 Line 219-222: Please remove the comma after “flow cytometric scoring”. That comma is not necessary.

Figure 2: Please add a plot displaying the MPO expression in granulocytes here too for completeness.

Reviewer #2: (No Response)

7. PLOS authors have the option to publish the peer review history of their article (what does this mean?). If published, this will include your full peer review and any attached files.

Reviewer #1: No

Reviewer #2: **Yes: **Valentina Giudice

---

## [Author Response · Author response to Decision Letter 1]

5 Sep 2022

Dear Prof. Shantsila,

We appreciate your decision to consider a second revision of our manuscript entitled “Comparative diagnostic accuracy between simplified and original flow cytometric gating strategies for peripheral blood neutrophil myeloperoxidase expression in ruling out myelodysplastic syndromes” (PONE-D-22-04816R1).

The funders had no role in study design, data collection and analysis, decision to publish, or preparation of the manuscript

We are very grateful to the reviewer for the thorough examination of our manuscript and for the practical presentation of his/her comments. Below, please find item-by-item responses to the reviewer’s comments, which are included verbatim. All page and paragraph numbers refer to locations in the revised manuscript.

We hope that the new version of our paper is deemed worthy of publication in PLOS ONE. 

Sincerely,

Tatiana Raskovalova 

Reviewer #1

1. Figure 1G: Following up on my last comment about the gating threshold in this plot, but now more specifically. What are the gray dots in this plot (I noticed these when I downloaded the Figure and looked very closely)? Are those MPO negative granulocytes and shouldn’t those be included in your analysis as well, as degranulation occurred in these granulocytes? Excluding granulocytes with low MPO expression levels creates a bias in your RCVs (although small in the displayed case). However, if there are more granulocytes with such low MPO expression levels, readers will exclude those based on the method displayed here, which is incorrect and will tremendously influence the resulting RCVs. To address this comment I would like to see the following: Please analyze the RCVs on all granulocytes without excluding any based on MPO expression levels. This will prevent any potential bias in the RCVs of MPO expression in the granulocytes.

We thank this Reviewer for the opportunity to clarify this point. Actually, RCV is robust to outliers and constitutes an alternative to the coefficient of variation for skewed distributions (See Bonett and Seier. Biomed J 2005 and Arachchige et al. J Appl Stat 2020). To address this Reviewer’s concern, we computed RCV based on all granulocytes (Panel A) and after excluding granulocytes with lower levels of MPO expression (Panel B), respectively. Reassuringly, the RCV value was 27.6 for both experiments. 

Panel A. All granulocytes Panel B. Excluding granulocytes with lower level of MPO expression

RCV = 27.6 RCV = 27.6

We agree with this Reviewer that the original gating strategy is somewhat subjective and operators with varying levels of expertise might introduce bias in estimating RCV values. In contrast, the gated cells are selected based on CD16 / CD11b double positivity, using the simplified gating strategy (Panel C). 

Panel C. Simplified gating strategy

RCV = 27.6

Changes in the revised manuscript in response to the above are highlighted below:

Simplified gating strategy (page 9):

“Mature neutrophils were individualized (CD16+ CD11b+) and gated cells were selected based on CD16/CD11b double positivity (D). The MPO SSC dot plot (E) was used only to visualize MPO expression of mature neutrophils.”

Myeloperoxidase expression (page 9):

“RCV is robust to outliers and constitutes an alternative to the coefficient of variation for skewed distributions.”

Discussion (page 20):

“Third, the original gating strategy was somewhat subjective and operators with varying levels of expertise might introduce bias in estimating RCV values. Reassuringly, the RCV value was rather unchanged after excluding granulocytes with lower levels of MPO expression.”

Minor

2. P5 Line 61: Please leave out Indeed in the beginning of the sentence.

This term has been removed from the revised manuscript.

3. P5 Line 68: This part of the sentence is not correctly written: “that constitutes the major component of neutrophil azurophilic granules”.

This sentence has been rephrased.

4. P5 Line 70: “MPO cytoplasmic expression” should be corrected to “”cytoplasmic expression of MPO”

This has been revised accordingly.

5. P5 Line 71: “immuno-cytochemical” should be corrected to “immunocytochemical”

This has been corrected.

6. P5 Line 76: Thank you for adding this passage to the manuscript. One small remark about part of this sentence: “relied on operator’s expertise”. Perhaps the authors can correct this to: “relied on the expertise of the operators”.

We have edited this sentence accordingly.

7. P6 Line 104: In this part of the sentence “we assessed agreement and comparative accuracy of RCV”, the abbreviation of RCV is first used and should be written in full with abbreviation between brackets. I also miss a word in the sentence. Please correct this part of the sentence to: “we assessed the agreement and comparative accuracy of the RCV.”

Actually, the first occurrence of RCV is located Page 5, line 87. The sentence has been corrected accordingly.

8. P7 Line 199-128: Please remove the bullet points and write this down in narrative form.

We have removed the bullet points from this para.

9. P10 Line 219-222: Please remove the comma after “flow cytometric scoring”. That comma is not necessary.

We have removed the comma from this sentence.

10. Figure 2: Please add a plot displaying the MPO expression in granulocytes here too for completeness.

We have modified Figure 2, as requested by this Reviewer.

---

## [Decision Letter · Decision Letter 2]

29 Sep 2022

Comparative diagnostic accuracy between simplified and original flow cytometric gating strategies for peripheral blood neutrophil myeloperoxidase expression in ruling out myelodysplastic syndromes.

PONE-D-22-04816R2

Dear Dr. Labarere,

We’re pleased to inform you that your manuscript has been judged scientifically suitable for publication and will be formally accepted for publication once it meets all outstanding technical requirements.

Kind regards,

Eduard Shantsila

Academic Editor

PLOS ONE

Additional Editor Comments (optional):

Reviewers' comments:

Reviewer's Responses to Questions

**Comments to the Author**

1. If the authors have adequately addressed your comments raised in a previous round of review and you feel that this manuscript is now acceptable for publication, you may indicate that here to bypass the “Comments to the Author” section, enter your conflict of interest statement in the “Confidential to Editor” section, and submit your "Accept" recommendation.

Reviewer #1: All comments have been addressed

Reviewer #2: All comments have been addressed

2. Is the manuscript technically sound, and do the data support the conclusions?

Reviewer #1: Yes

Reviewer #2: Yes

3. Has the statistical analysis been performed appropriately and rigorously? 

Reviewer #1: Yes

Reviewer #2: Yes

4. Have the authors made all data underlying the findings in their manuscript fully available?

Reviewer #1: Yes

Reviewer #2: Yes

5. Is the manuscript presented in an intelligible fashion and written in standard English?

Reviewer #1: (No Response)

Reviewer #2: Yes

6. Review Comments to the Author

Reviewer #1: I would like to thank the authors for taking the time and effort to address my comments and concerns. I think that the manuscript is acceptable for publication now.

However, I would like to stress that a reduction in MPO expression is a normal phenomenon in MDS (due to hypogranulation as you probably know). Granulocytes with low MPO expression are thus biologically relevant in MDS cases and it sounds questionable to remove such cells from the analysis. As the new simplified gating strategy does not include removal of the granulocyes with low MPO expression, this concern is readily addressed.

Reviewer #2: (No Response)

7. PLOS authors have the option to publish the peer review history of their article (what does this mean?). If published, this will include your full peer review and any attached files.

Reviewer #1: No

Reviewer #2: No

---

## [Editor Report · Acceptance letter]

10 Nov 2022

PONE-D-22-04816R2 

Comparative diagnostic accuracy between simplified and original flow cytometric gating strategies for peripheral blood neutrophil myeloperoxidase expression in ruling out myelodysplastic syndromes. 

Dear Dr. Raskovalova:

I'm pleased to inform you that your manuscript has been deemed suitable for publication in PLOS ONE. Congratulations! Your manuscript is now with our production department. 

Kind regards, 

on behalf of

Dr. Eduard Shantsila 

Academic Editor

PLOS ONE